# Simulation of the Cyclic Stress–Strain Behavior of the Magnesium Alloy AZ31B-F under Multiaxial Loading

Vitor Anes [1,2,3,*], Rogério Moreira [3], Luís Reis [3] and Manuel Freitas [3]

1    Instituto Superior de Engenharia de Lisboa, 1959-007 Lisboa, Portugal
2    Instituto Politécnico de Lisboa, 1549-020 Lisboa, Portugal
3    IDMEC, Instituto Superior Técnico, Universidade de Lisboa, 1049-001 Lisboa, Portugal; a46612@alunos.isel.pt (R.M.); luis.g.reis@tecnico.ulisboa.pt (L.R.); manuel.freitas@tecnico.ulisboa.pt (M.F.)
*    Correspondence: vitor.anes@isel.pt

**Abstract:** Under strain control tests and cyclic loading, extruded magnesium alloys exhibit a special mechanism of plastic deformation ("twinning" and "de-twining"). As a result, magnesium alloys exhibit an asymmetric material behavior that cannot be fully characterized with the typical numerical tools used for steels or aluminum alloys. In this sense, a new phenomenological model, called hypo-strain, has been developed to correctly predict the cyclic stress–strain evolution of magnesium alloys. On this basis, this work aims to accurately describe the local cyclic elastic–plastic behavior of AZ31B-F magnesium alloy under multiaxial cyclic loading with Abaqus incremental plasticity. The phenomenological hypo-strain model was implemented in the UMAT subroutine of Abaqus/Standard to be used as a design tool for mechanical design. To evaluate this phenomenological approach, the results were correlated with the uniaxial and multiaxial proportional and non-proportional experimental tests. In addition, the estimates were also correlated with the Armstrong–Frederick nonlinear kinematic hardening model. The results show a good correlation between the experiments and the phenomenological hypo strain approach. The model and its implementation were validated in the strain range studied.

**Keywords:** AZ31B-F; magnesium alloys; multiaxial loading; simulation; cyclic stress–strain behavior





## 1. Introduction

Environmental concerns have led to the reduction of pollution and fuel consumption becoming an important goal of the transportation industry [1,2]. Although magnesium alloys are mainly used in non-structural castings, wrought magnesium alloys are increasingly used in structural applications [3–6]. Magnesium alloys (MA) are attractive for use in structural components due to their good strength-to-weight ratio, low density, and high damping capacity [7–10].

In general, components and structures are subjected to cyclic multiaxial loading, so the study of the multiaxial fatigue behavior of MA is an important topic. Although these components are expected to operate in a high cycle range, the study of fatigue at low cycles is important, because local plasticity can occur at geometric irregularities or notches [11–13]. Due to stress concentrations (notches, material defects, or surface roughness), the local material yields first to redistribute the load to the surrounding material after cyclic plastic deformation. In these situations, finite element analysis allows a better understanding of the evolution of local cyclic stresses and strains [14,15].

Materials tend to change their mechanical properties, such as strength, with the type and amount of loading. In addition, the number of slip planes has been found to have a great influence on the cyclic behavior of materials. MA has a hexagonal closed microstructure (HCP) with three basal slip planes, and it is necessary to activate non-basal slip planes or other deformation modes, such as twinning, to show good ductility.

Under cyclic loading, extruded magnesium alloys exhibit certain plastic deformation mechanisms (sliding, twinning, and de-twinning) that cause the asymmetric material behavior [16–18]. In recent years, extensive strain-controlled uniaxial fatigue tests have been performed on wrought magnesium alloys [19–27]. However, there are few works in the literature on multiaxial fatigue of magnesium alloys [27–33]. Zhang et al. [29] performed combined axial-torsional fatigue tests on AZ61A magnesium alloy in thin-walled tubular specimens. They found that the alternating occurrence of twinning and de-twinning processes under axial loading affects the shear component, resulting in asymmetric hysteresis loops between shear stress and shear strain. Xiong et al. [30] investigated the multiaxial fatigue behavior of an extruded AZ31B magnesium alloy. It was found that the non-proportional loading had a detrimental effect on the fatigue life compared to the proportional loading. The modified Smith–Watson–Topper and Jiang models were able to estimate the multiaxial fatigue life. Jahed et al. [31] compared two extruded magnesium alloys, AZ31B and AZ61A, under multiaxial fatigue. They found that these alloys have similar properties and cyclic mechanical behavior.

Most of these uniaxial and multiaxial studies focus on fatigue crack initiation and propagation, fatigue life estimation, and mechanical behavior characterization. However, there are only a limited number of published studies on modeling the cyclic stress–strain evolution of MA [34–37]. Lee et al. [35] developed a constitutive plasticity model to capture the anisotropic and asymmetric behavior of MA under uniaxial tension–compression loading. They extended a strain hardening law based on two-surface models to account for the Bauschinger effect, transient behavior, and the unusual asymmetry of MA. The Drucker–Prager yield surface was modified to account for the anisotropy of MA. Li et al. [36] used a von Mises yield function with a nonzero initial back stress with isotropic and nonlinear kinematic hardening to account for the asymmetry of plastic flow in MA. Both constitutive plasticity models were implemented in Abaqus via subroutines and developed for MA plates. The formulations were derived from the plane stress conditions, which limit the applicability of the models. Although these models showed good agreement with experimental tests for uniaxial fatigue, they were not tested for multiaxial fatigue.

The constitutive plasticity models have three main functions for estimating the stress–strain relationships for each type of loading. These functions are the yield function, the yield rule, and the hardening rule, which are based on a single loading path, usually under monotonic or cyclic uniaxial loading conditions [38–43]. However, when multiaxial loading is considered, the yield function, the hardening rule, and the yield rule vary in different ways, depending on the type and magnitude of the loading [32,44].

In addition, experiments have shown that the severity of the asymmetric strain hardening behavior of magnesium alloys depends on several factors, including the magnitude of the load and the type of load [29,32,33]. The vast majority of constitutive plasticity models found in the literature are based on a single reference loading case, which limits their accuracy in describing the multiaxial stress–strain hysteresis loops of MA for different loading paths and levels.

A new phenomenological approach, called hypo-strain, has been developed to capture the asymmetric behavior of magnesium alloys [45–47]. This phenomenological model is a mapping of the cyclic response of the material to MA, based on experimental tests. Mapping the cyclic elastic–plastic behavior of MA is a feasible way of capturing the particular plastic deformation of MA, such as twinning, de-twinning, and the slip effect, at any strain magnitude and loading mode.

In the present work, this new phenomenological model has been implemented in the UMAT subroutine of Abaqus/Standard. This work is the first attempt to develop a numerical tool that allows the designer to accurately describe the local cyclic stress–strain evolution of wrought MA. The finite element analyses are crucial for studying the interactions between the different material elements during elastic–plastic deformations and stress redistribution. In addition, these estimates can be used as input to fatigue damage models to improve the accuracy of life prediction. In this work, the phenomenological

hypo-strain model implemented in the Abaqus UMAT subroutine was evaluated for the AZ31B-F magnesium alloy under multiaxial fatigue conditions.

## 2. Materials and Methods

### 2.1. Theoretical Models

This section presents the theoretical models used in the simulations. The Armstrong–Frederick plasticity model included in Abaqus and the hypo-strain model developed by the present authors are presented. The goal is to use these two models in the simulations and then correlate the results with the experimental data. At the end of this section, the experimental AZ31B-F data using the hypo-strain model are presented.

#### 2.1.1. Armstrong–Frederick Plasticity Model

The Armstrong–Frederick model is a nonlinear kinematic model with a yield function F given by Equation (1):

$$F = \sqrt{\frac{3}{2}\left(s - \alpha^{dev}\right) : \left(s - \alpha^{dev}\right)} - k = 0 \tag{1}$$

where $s$ is the deviatoric stress, $\alpha$ is the back stress, and $k$ is the yield stress. The kinematic hardening is governed through the back stress tensor that can be calculated with Equation (2):

$$d\alpha = \frac{2}{3}Cd\varepsilon_p - \gamma\alpha dp \tag{2}$$

where $C$ and $\gamma$ are material parameters, $d\alpha$ is the increment of the back stress, and $d\varepsilon_p$ is the increment of plastic strain. The quantity $dp$ is the increment of the accumulated plastic strain. In Equation (2), the material parameter $C$ expresses the hardening modulus and $\gamma$ controls the rate at which the hardening modulus decreases with the increase in plastic strain. Armstrong–Frederick material parameters were determined through uniaxial cyclic stress–strain tests under stabilized hysteresis loops. Different strain amplitudes of the strain-controlled tests were used to calibrate the material parameters ($C$ and $\gamma$). The uniaxial cyclic stress–strain curves of AZ31B-F magnesium alloy present different values for the yield stress in tension and compression, with the tensile stress higher than compression. Therefore, as a conservative approach to the Armstrong–Frederick plasticity model, the tensile branch of the AZ31B-F magnesium alloy cyclic response was considered. Having the plastic strain amplitude values ($\Delta\varepsilon_{pl}/2$) and the inherent recall term, which is the difference between the stress amplitude ($\Delta\sigma/2$) applied and the material cyclic yield stress ($k$), for a fixed total strain, the relation between these two values can be fitted with Equation (3):

$$\frac{\Delta\sigma}{2} - k = \frac{C}{\gamma}\tanh\left(\gamma\frac{\Delta\varepsilon_{pl}}{2}\right) \tag{3}$$

#### 2.1.2. Hypo-Strain Phenomenological Model

This new phenomenological model is a representation of the material behavior of wrought magnesium alloys based on a variety of experimental tests. This approach uses third-degree polynomial functions to capture the fully developed behavior of these materials. Experimental tests have shown that MA exhibits asymmetric strain hardening behavior with different yield stresses in compression and tension. In addition, the back stresses and plastic strains are different in tension and compression. Therefore, to obtain the polynomial functions described in [45–47], six specific points on a stabilized hysteresis loop are considered, as shown in Figure 1.

For the tension branch, the polynomial function is determined using the experimental data at points 4, 5, 6, and 1. Similarly, for the compression branch, the polynomial function is determined using the experimental data at points 4, 3, 2, and 1. This procedure is repeated for the shear hysteresis loop. The functions $P1(\varepsilon_{sl}, \lambda)$ and $P4(\varepsilon_{sl}, \lambda)$ estimate the stresses for the maximum total strain, and for positive and negative loading directions, respectively.

Functions $P2(\varepsilon_{sl}, \lambda)$ and $P5(\varepsilon_{sl}, \lambda)$ estimate the plastic strain increments associated with the maximum total strain, and functions $P3(\varepsilon_{sl}, \lambda)$ and $P6(\varepsilon_{sl}, \lambda)$ estimate the back stresses. Under biaxial tension–torsion loading conditions, two hysteresis loops can be obtained, one concerning the axial loading component and another one for the shear loading component; both are dependent on each other. This dependence is captured by the strain amplitude ratio ($\lambda = \gamma/\varepsilon$) given by the shear strain ($\gamma$) to axial strain ($\varepsilon$) ratio. The biaxial strain magnitude is given by: $\varepsilon_{sl} = \sqrt{\varepsilon_t^2 + \gamma_t^2}$. These functions, $P1(\varepsilon_t, \lambda)$ to $P6(\varepsilon_t, \lambda)$, vary according to the strain amplitude ratio ($\lambda$) and the strain magnitude ($\varepsilon_t$).

With these polynomials, it is possible to capture the mechanisms of plastic deformation of magnesium alloys, such as twinning, de-twinning, and slip effects, at any strain level and strain amplitude ratio.

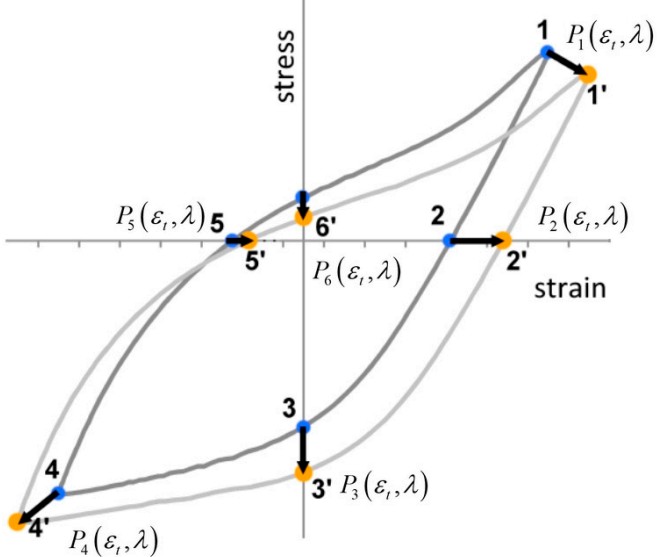

**Figure 1.** Third degree polynomial interpolation for two hysteresis loops (axial and shear).

The axial and shear hysteresis loops for a given total strain are given by Equation (4) for the axial stress component and by Equation (5) for the shear component.

$$\sigma_{tension}(\varepsilon_t) = a_{\varepsilon_t}\varepsilon_t^3 + b_{\varepsilon_t}\varepsilon_t^2 + c_{\varepsilon_t}\varepsilon_t + d_{\varepsilon_t}$$
$$\sigma_{compresion}(\varepsilon_t) = e_{\varepsilon_t}\varepsilon_t^3 + f_{\varepsilon_t}\varepsilon_t^2 + g_{\varepsilon_t}\varepsilon_t + h_{\varepsilon_t} \tag{4}$$

where $a_{\varepsilon t}$, $b_{\varepsilon t}$, $c_{\varepsilon t}$, and $d_{\varepsilon t}$ are the polynomial coefficients for the tension branch of the hysteresis loop and $e_{\varepsilon t}$, $f_{\varepsilon t}$, $g_{\varepsilon t}$, and $h_{\varepsilon t}$ are the coefficients for the compression branch. In the same equation, $\varepsilon_t$ is the axial total strain.

$$\tau_{+direction}(\gamma_t) = a_{\gamma_t}\gamma_t^3 + b_{\gamma_t}\gamma_t^2 + c_{\gamma_t}\gamma_t + d_{\gamma_t}$$
$$\tau_{-direction}(\gamma_t) = e_{\gamma_t}\gamma_t^3 + f_{\gamma_t}\gamma_t^2 + g_{\gamma_t}\gamma_t + h_{\gamma_t} \tag{5}$$

Similarly, $a_{\gamma t}$, $b_{\gamma t}$, $c_{\gamma t}$, and $d_{\gamma t}$ are the polynomial coefficients for the positive direction branch and $e_{\gamma t}$, $f_{\gamma t}$, $g_{\gamma t}$, and $h_{\gamma t}$ are the coefficients for the negative direction branch. In the same equation, $\gamma_t$ is the shear total strain.

In this study, the polynomial constants of Equations (4) and (5) were obtained with a Matlab polyfit function, which has, as input values, the output values of the functions $P1(\varepsilon_{sl}, \lambda 1)$ to $P6(\varepsilon_{sl}, \lambda 1)$. These functions can be obtained with Equations (6) and (7). Based on experiments, these functions have the following shape under multiaxial loading conditions:

$$P_{axial,n}(\varepsilon_{sl}, \lambda) = a_i + b_i\varepsilon_{sl} + c_i\lambda + d_i\varepsilon_{sl}^2 + e_i\lambda^2 + f_i\varepsilon_{sl}\lambda + g_i\varepsilon_{sl}^3 + h_i\lambda^3 + i_i\varepsilon_{sl}\lambda^2 + j_i\varepsilon_{sl}^2\lambda \quad (6)$$

$$P_{shear,n}(\varepsilon_{sl}, \lambda) = a_j + b_j\varepsilon_{sl} + c_j\lambda + d_j\varepsilon_{sl}^2 + e_j\lambda^2 + f_j\varepsilon_{sl}\lambda + g_j\varepsilon_{sl}^3 + h_j\lambda^3 + i_j\varepsilon_{sl}\lambda^2 + j_j\varepsilon_{sl}^2\lambda \quad (7)$$

where $(a_i, b_i, c_i, \ldots, j_i)$ are coefficients for the axial component and $(a_j, b_j, c_j, \ldots, j_j)$ are the coefficients for the shear component of the bi-dimensional third-degree polynomial. $\varepsilon_{sl}$ is the strain magnitude and $\lambda$ is the strain amplitude ratio. The subscripts $i$ and $j$ denote the axial and shear components, respectively.

The subscript $n = 1$ to 6 represents the six specific points of the hysteresis loop, as described in Figure 1. The constants of the two-dimensional polynomial of degree three were determined using the *polyfit* routine available in the Matlab Toolbox, which provides the coefficients for a polynomial p(x) of degree $n$ that better fits the data. These constants are based on the experimental results of the AZ31B-F magnesium alloy, and are given in Table 1 for the axial component and in Table 2 for the shear component.

**Table 1.** Polynomial constants for the $P$ functions of the axial component.

| Coefficients | $P_{axial,1}$ [MPa] | $P_{axial,2}$ [%] | $P_{axial,3}$ [MPa] | $P_{axial,4}$ [MPa] | $P_{axial,5}$ [%] | $P_{axial,6}$ [MPa] |
|---|---|---|---|---|---|---|
| $a_i$ | 267.7 | $-0.313$ | 170 | $-182.4$ | $-0.074$ | 3.311 |
| $b_i$ | $-1194$ | 2.54 | $-1041$ | $-104.1$ | 1.274 | $-492.9$ |
| $c_i$ | 0.029 | $-0.002$ | $-1.613$ | 4.79 | $-0.005$ | 4.498 |
| $d_i$ | 2923 | $-5.096$ | 1818 | 91.56 | $-3.02$ | 1424 |
| $e_i$ | 0.135 | $-2.57 \times 10^{-4}$ | 0.132 | $-0.043$ | $-3.32 \times 10^{-5}$ | $-0.023$ |
| $f_i$ | $-11.17$ | 0.027 | $-5.298$ | $-6.508$ | 0.01 | $-10.3$ |
| $g_i$ | $-1606$ | 2.97 | $-959.7$ | $-15.73$ | 1.341 | $-788.4$ |
| $h_i$ | $-0.001$ | $3.54 \times 10^{-6}$ | $-0.001$ | $3.09 \times 10^{-4}$ | $7.90 \times 10^{-7}$ | $-5.80 \times 10^{-5}$ |
| $i_i$ | 0.058 | $-2.40 \times 10^{-4}$ | 0.07 | 0.047 | $-3.15 \times 10^{-5}$ | 0.047 |
| $j_i$ | 1.952 | $-0.007$ | $-0.262$ | 1.817 | $8.92 \times 10^{-4}$ | 2.8 |

**Table 2.** Polynomial constants for the $P$ functions of the shear component.

| Coefficients | $P_{shear,1}$ [MPa] | $P_{shear,2}$ [%] | $P_{shear,3}$ [MPa] | $P_{shear,4}$ [MPa] | $P_{shear,5}$ [%] | $P_{shear,6}$ [MPa] |
|---|---|---|---|---|---|---|
| $a_j$ | $-10.57$ | $-2.232$ | $-259.6$ | 1334 | $-1.633$ | 79.78 |
| $b_j$ | $-223.4$ | 11.15 | 1150 | $-6194$ | 8.208 | $-195.7$ |
| $c_j$ | 3.489 | 0.008 | 2.656 | $-10.98$ | 0.007 | $-2.986$ |
| $d_j$ | 348.3 | $-17.08$ | $-1847$ | 9637 | $-12.97$ | 352.2 |
| $e_j$ | $-0.051$ | $-5.18\ 10^{-5}$ | $-0.04$ | 0.144 | $-6.90 \times 10^{-5}$ | 0.055 |
| $f_j$ | 3.671 | $-0.026$ | $-2.455$ | 10.22 | $-0.02$ | 1.288 |
| $g_j$ | $-171.3$ | 8.434 | 956.5 | $-4815$ | 6.536 | $-191.2$ |
| $h_j$ | $2.62 \times 10^{-4}$ | $-1.55 \times 10^{-5}$ | $1.58 \times 10^{-4}$ | $-4.56 \times 10^{-4}$ | $-2.06 \times 10^{-7}$ | $2.70 \times 10^{-4}$ |
| $i_j$ | $-0.016$ | $1.59 \times 10^{-4}$ | 0.031 | $-0.113$ | $2.29 \times 10^{-4}$ | $-0.015$ |
| $j_j$ | $-1.27$ | 0.014 | $-1.05$ | $-0.798$ | $-0.005$ | 0.31 |

### 2.2. Methods

The implementation of the phenomenological hypo-strain model in the finite element software (FE) required a nonstandard approach, since this analytical model is based on third degree polynomial functions. In addition, this model does not follow any specific yield function, yield rule, and hardening rule used in constitutive plasticity models. Therefore, the implementation of this model in the UMAT subroutine is based on updating the tangential stiffness matrix during cyclic loading for different stress states, according to the triangular and sinusoidal wave displacement amplitudes. This new approach allows an accurate description of the stress–strain evolution of AZ31B-F magnesium alloys during uniaxial and multiaxial fatigue. UMAT is universal; it can be used with any type of loading,

proportional and non-proportional. The mapping of the cyclic loading response of the material takes into account the experimental loads and estimates the material response for loads that were not considered in the experiments. In this sense, this section provides a detailed description of the numerical implementation of the hypo-strain model in the UMAT subroutine of Abaqus. In addition, this section provides a detailed description of the work performed on this topic, as well as the pre-processing, processing, and post-processing features of Abaqus.

### 2.2.1. Numerical Simulation Setup

The numerical simulations focused on the evolution of the cyclic stresses/strains under uniaxial and multiaxial loading. To perform the numerical analysis, the thin-walled specimen used in the experiments (Figure 2a) was restricted to a specimen with constant cross-sectional area, as shown in Figure 2b. The specimen used for the numerical simulation has an outer diameter of 10 mm, an inner diameter of 8 mm, and a length of 25 mm. Symmetrical conditions were considered in the simulations to reduce the computation time. This specimen was selected for the simulations for two reasons. One was that the results obtained in the experiments were obtained with a similar constant cross-sectional area. The other reason was to simplify the implementation of the phenomenological model in the UMAT subroutine of Abaqus.

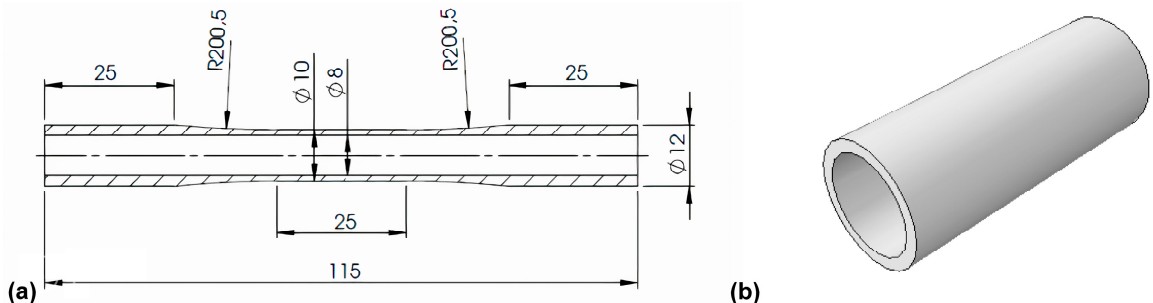

**Figure 2.** (**a**) specimen used on the experiments; (**b**) specimen used on numerical simulations.

The phenomenological model was later correlated with the well-known Armstrong–Frederick model. In the simulations, the nonlinear kinematic hardening model was used, instead of the nonlinear combined isotropic/kinematic model for comparison with the phenomenological approach. The reason for this is that this study aims to describe the stabilized hysteresis loops of MA. The materials reach the stabilized cyclic deformations after initial transient cyclic deformations.

The AZ31B-F experiments were performed under strain control with different strain levels for the 6 loading paths shown in Figure 3. The first loading case, case 1, is a pure uniaxial push-pull test, named PT. The second case, named case 2, is a pure shear loading, and is referred to as case PS. Cases 3, 4, and 5 are proportional loads with strain amplitude ratios for the 30°, 45°, and 60° conditions, respectively, and which are referred to as PP30, PP45, and PP60, respectively. Finally, Case 6, named OP45, is a non-proportional loading case with a strain amplitude ratio for the 45° strain amplitude ratio and a phase shift of 90°. The strain results were measured using a biaxial strain transducer on a similar section of the specimen. The phenomenological model, implemented via a UMAT subroutine of Abaqus/Standard [48], was validated against a variety of strain paths considering shear ($\gamma$) and normal ($\varepsilon$) strain planes, as shown in Figure 3. This figure shows the six biaxial strain paths that were implemented.

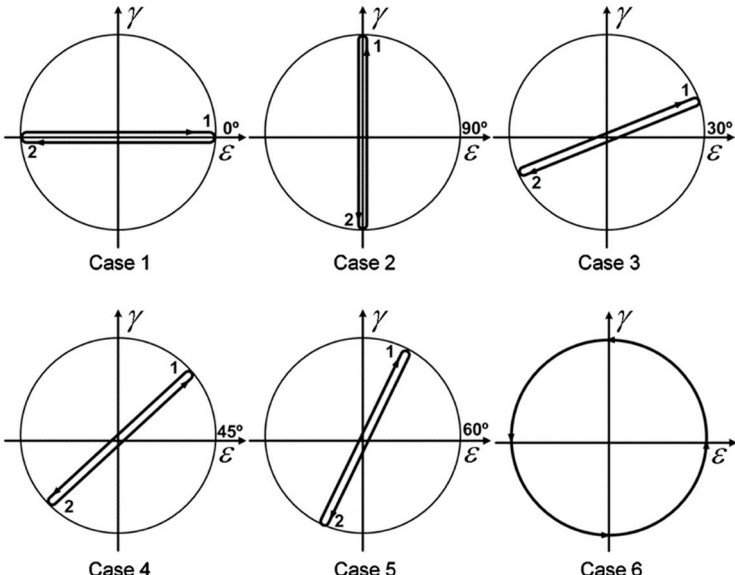

**Figure 3.** Loading paths performed in strain control: Case 1—PT, Case 2—PS, Case 3—PP30, Case 4—PP45, Case 5—PP60, and Case 6—OP45.

In the numerical simulations, an isoparametric hexahedral body with linear 8-node brick elements was used. First-order interpolation was used for these types of elements. The FE analyses were performed under quasi-static conditions with implicit time integration. The implicit procedure solves the equilibrium equations at each time step. This procedure ensures the accuracy and stability of the analysis. Considering the lower computational time of the simulations, a full model with full integration algorithm was used. This algorithm has the advantage of preventing hourglass modes that can occur when the tangential stiffness matrix is computed via the UMAT subroutine of Abaqus and reducing mesh distortions. A swept mesh technique with a medial axis provides good uniformity and improves the quality of the mesh. A convergence study was performed to ensure that the number of mesh elements does not affect the results. The final mesh contains 5800 elements and 8874 nodes. Figure 4 shows the thin-walled specimen with C3D8 mesh elements used for the numerical simulations. These elements are general purpose linear brick elements. As boundary conditions for the tensile-torsional loading, the specimen was fixed at one end and displacement about the longitudinal axis of the specimen and rotation about the same axis were performed at the other end, at the reference point. In the numerical simulations, a kinematic coupling condition was used to bind the nodes at the end of the specimen surface to the rigid-body motion of the reference point.

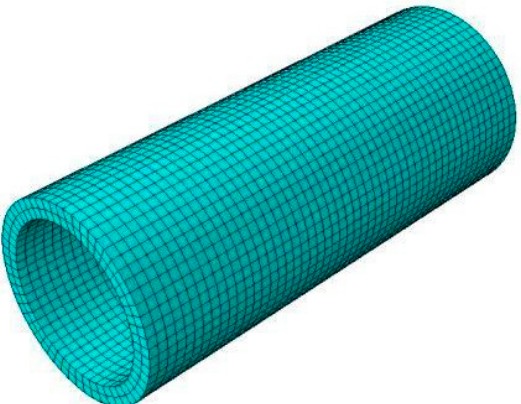

**Figure 4.** Mesh model representation of the specimen used on numerical simulations.

### 2.2.2. User-Defined Material Model

The phenomenological approach was implemented in the FE Abaqus/Standard software package via the UMAT subroutine. Three tasks were required to implement the phenomenological model: (1) defining the material parameters; (2) calculating the tangent stiffness matrix; (3) updating the material parameters and the tangent stiffness matrix based on different stress states.

Definition of material parameters. In the analytical phenomenological approach, the hysteresis loops of AZ31B-F magnesium alloy were approximated by third degree polynomial functions for arbitrary values of total strains. For each hysteresis loop, two polynomials were used, one for the positive direction branch and another for the negative one. Based on the polynomial functions, the parameters were defined: $E_{ep}(\lambda, \varepsilon_{sl})$ and $G_{ep}(\lambda, \varepsilon_{sl})$. These material parameters were used to update the stress–strain cyclic relation according to the strain amplitude ratio ($\lambda$) and strain magnitude ($\varepsilon_{sl}$). These two material parameters aim to map the asymmetric material response of magnesium alloys under uniaxial and multiaxial fatigue. Considering the previous Equation (4), the $E_{ep}$ material parameter was defined in function of total axial strain, such as in Equation (8):

$$E_{ep} = a_{\varepsilon_t}\varepsilon_t^2 + b_{\varepsilon_t}\varepsilon_t + c_{\varepsilon_t} + \frac{d_{\varepsilon_t}}{\varepsilon_t} \tag{8}$$

Where $a_{\varepsilon t}$, $b_{\varepsilon t}$, $c_{\varepsilon t}$, and $d_{\varepsilon t}$ are the axial polynomial coefficients of the tension branch and $\varepsilon_t$ is the axial total strain. The material parameter $E_{ep}$ corresponds to elastic–plastic modulus. If the material is in elastic domain, it expresses the Young modulus. If the material is in plastic domain it expresses the tangent modulus. Similarly, considering Equation (5), the $G_{ep}$ material parameter was defined in the function of total shear strain as Equation (9):

$$G_{ep} = a_{\gamma_t}\gamma_t^2 + b_{\gamma_t}\gamma_t + c_{\gamma_t} + \frac{d_{\gamma_t}}{\gamma_t} \tag{9}$$

Similarly, $a_{\gamma t}$, $b_{\gamma t}$, $c_{\gamma t}$, and $d_{\gamma t}$ are the shear polynomial coefficients of the positive direction branch and $\gamma_t$ is the total shear strain. $G_{ep}$ is another material parameter and corresponds to shear elastic–plastic modulus. Similarly, it expresses the elastic shear modulus in the elastic domain and the elastic–plastic shear modulus in the plastic domain. This procedure is repeated for the compressive branch of the axial component and for the branch of the negative direction of the shear component. Definition of the tangential stiffness matrix. Considering the elastic–plastic behavior and the total strain increment, the stress increment is obtained from Equation (10):

$$d\sigma = Cd\varepsilon = C(d\varepsilon^e + d\varepsilon^p) \tag{10}$$

where $C$ is the tangent stiffness matrix and $d\varepsilon$, $d\varepsilon^e$, and $d\varepsilon^p$ are the total, elastic, and plastic strain increment, respectively. For a given total strain increment $d\varepsilon$, prescribed at every time increment, the stress is updated with Equation (10). In the present work, no decomposition of the total strain increment into elastic and plastic increment was carried out. The phenomenological model was implemented in UMAT, based on the definition of the tangent stiffness matrix. In this study, the fourth-order tangent stiffness matrix was defined based on two material parameters, $E_{ep}(\lambda, \varepsilon_{sl})$ and $G_{ep}(\lambda, \varepsilon_{sl})$, which are a function of the strain magnitude and loading path, and based on the Poisson ratio.

Although the extruded MA also exhibits anisotropic material behavior, this phenomenological model implemented in UMAT aims to only capture the asymmetric behavior of magnesium alloys. Therefore, the following isotropic tensor was defined, referred to here as the discrete stiffness matrix $C^*$ and represented by Equation (11).

$$C^* = \begin{bmatrix} \frac{(1-\nu)E_{ep}(\lambda,\varepsilon_{sl})}{(1+\nu)(1-2\nu)} & \frac{\nu E_{ep}(\lambda,\varepsilon_{sl})}{(1+\nu)(1-2\nu)} & \frac{\nu E_{ep}(\lambda,\varepsilon_{sl})}{(1+\nu)(1-2\nu)} & 0 & 0 & 0 \\ \frac{\nu E_{ep}(\lambda,\varepsilon_{sl})}{(1+\nu)(1-2\nu)} & \frac{(1-\nu)E_{ep}(\lambda,\varepsilon_{sl})}{(1+\nu)(1-2\nu)} & \frac{\nu E_{ep}(\lambda,\varepsilon_{sl})}{(1+\nu)(1-2\nu)} & 0 & 0 & 0 \\ \frac{\nu E_{ep}(\lambda,\varepsilon_{sl})}{(1+\nu)(1-2\nu)} & \frac{\nu E_{ep}(\lambda,\varepsilon_{sl})}{(1+\nu)(1-2\nu)} & \frac{(1-\nu)E_{ep}(\lambda,\varepsilon_{sl})}{(1+\nu)(1-2\nu)} & 0 & 0 & 0 \\ 0 & 0 & 0 & G_{ep}(\lambda,\varepsilon_{sl}) & 0 & 0 \\ 0 & 0 & 0 & 0 & G_{ep}(\lambda,\varepsilon_{sl}) & 0 \\ 0 & 0 & 0 & 0 & 0 & G_{ep}(\lambda,\varepsilon_{sl}) \end{bmatrix} \qquad (11)$$

Updating the material parameters and the discrete stiffness matrix. The material parameters $E_{ep}(\lambda, \varepsilon_{sl})$ and $G_{ep}(\lambda, \varepsilon_{sl})$ and the discrete stiffness matrix were updated in ten (t1 to t10 described in Figure 5) different stress states during cyclic loading according to the triangular displacement amplitude, as shown in Figure 5, except for the nonproportional loading. For the non-proportional loading, the amplitude of the sinusoidal waveform updated in twelve different stress states was used to accurately describe the stress–strain hysteresis loops of AZ31B-F magnesium alloy. In this work, the following considerations were taken into account:

- Poisson's ratio was set equal to 0.35 and considered constant for elasticity and plasticity.
- The parameters $E_{ep}(\lambda, \varepsilon_{sl})$ and $G_{ep}(\lambda, \varepsilon_{sl})$, as well as the discrete stiffness matrix, are valid only in one stress state during cyclic loading;
- Between two successive stress states, the material parameters ($E_{ep}$ and $G_{ep}$) were assumed to vary linearly.

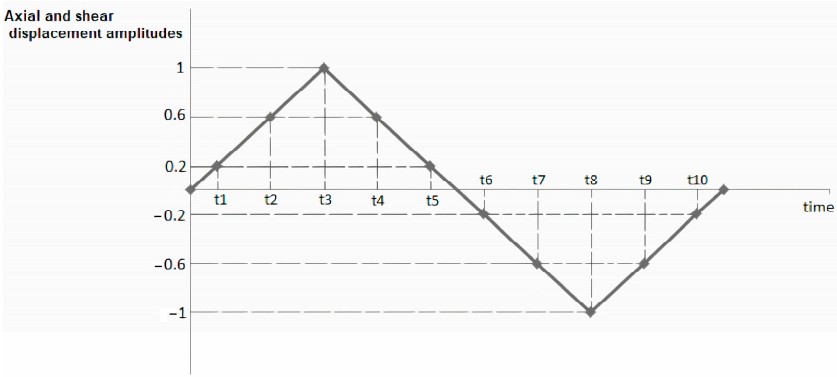

**Figure 5.** Triangular displacement amplitude.

### 2.2.3. Implementation Issues in the UMAT Subroutine of Abaqus

To run the simulations on Abaqus/Standard 6.14, integration of Visual Studio 2012 and Intel Visual FORTRAN Composer XE 2013 was required. The user must have these programs linked together to run the UMAT subroutine.

Figure 6 shows the data flow of the hypo-strain implementation model on ABAQUS and the interactions between the preprocessing, processing, and postprocessing information. The numerical implementation of the phenomenological model consisted of the creation of two files: the INPUT file (.inp) and the FORTRAN file (.for). The INPUT file contains all the input data needed to run the simulation, the model data and the history data. These data define the sample and boundary conditions used for the low cycle fatigue simulations of the AZ31B-F magnesium alloy. This file is the means of communication between the preprocessor (ABAQUS/CAE) and the analysis product (ABAQUS/Standard), and contains the input data required by ABAQUS/Standard. In the same way, the FORTRAN file contains the code for the phenomenological hypo-strain stretching model needed to define the UMAT subroutine. This information is sent to Abaqus/Standard, which solves the nonlinear finite element equilibrium equations. During the analysis, Abaqus/Standard sends information to Abaqus/CAE so that the user can monitor the progress of the computational

simulation. After convergence is achieved and the quasi-static analysis is successfully completed, the results can be viewed in the output database (.odb).

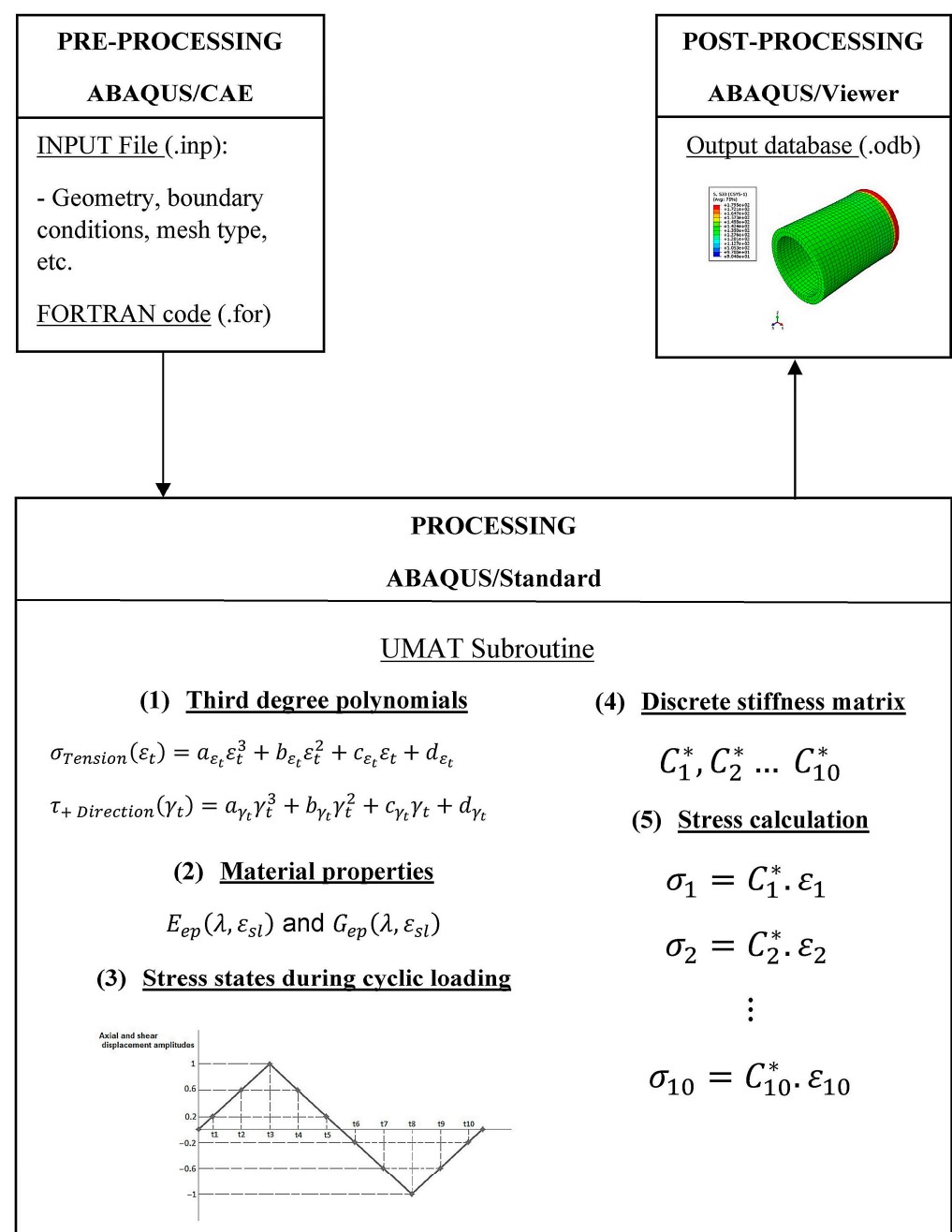

**Figure 6.** Data flow of the hypo-strain phenomenological implementation on ABAQUS.

## 3. Results

### 3.1. Armstrong–Frederick Plasticity Calibration Results

The Matlab fitting toolbox based on uniaxial stress–strain experimental tests was used to obtain the material constants $C$ = 32,736 MPa and $\gamma_{AF}$ = 211. Figure 7 shows the data fitting curve obtained for the AZ31B-F magnesium alloy.

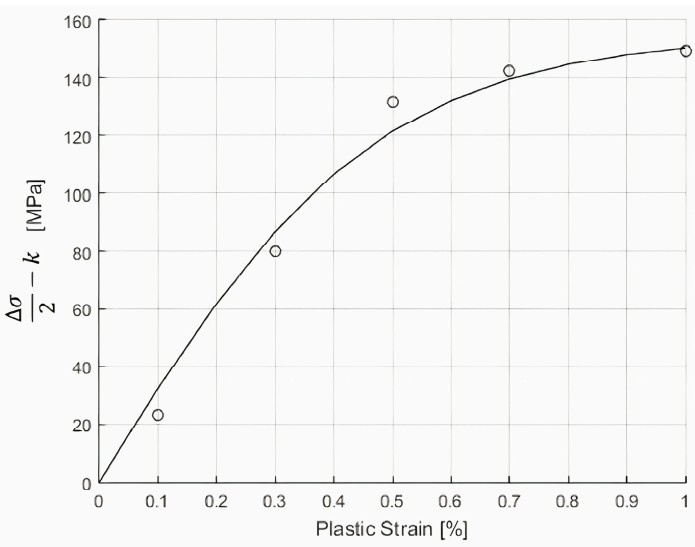

**Figure 7.** Armstrong–Frederick plasticity model calibration, correlation factor $R^2$ = 0.92.

### 3.2. Pure Axial (Case PT)

From the experimental results of pure axial loading—shown in Figure 8, with strain amplitudes of 0.5% and 1.2%—it can be concluded that the hysteresis loops show asymmetric behavior. However, this is more evident at 1.2% of the total strain. When the tensile part of the hysteresis loops is compared with the compressive part, the asymmetry becomes clear: the plastic strains and the back stresses (which are the stresses required to reduce the plastic strains to zero) are different in tension and compression. In addition, the axial stress for the maximum amplitude of the total strain is different from that of compression. The results show the presence of six independent parameters in the hysteresis loops of magnesium alloy AZ31B-F, as described in Figure 1. These parameters are independent and are accounted for in the phenomenological hypo-strain model by Equation (6) for the axial component and by Equation (7) for the shear component. From the uniaxial results shown in Figure 8, it can be concluded that the phenomenological model follows the experimental results with very good accuracy, except for the purely axial loading with 1.2% of the total strain, where a conservative deviation is observed during compressive loading. It is also found that the phenomenological model implemented in the UMAT subroutine of Abaqus is very close to the analytical approach. As for the Armstrong–Frederick model, the only estimate that has acceptable accuracy is that for pure axial loading with 0.5% of the total strain. For pure axial loading at 1.2% of total strain, the Armstrong–Frederick model provides a good estimate for plastic strain and maximum stress limit in tension, but overestimates the maximum stress limit in compression and back stress. In the axial-only test with a total strain of 1.2%, an inflection point was observed on the stress–strain curves, followed by a sudden increase in strain hardening behavior during tensile loading. This can be attributed to the peculiar mechanism of cyclic plastic deformation of magnesium alloys (twinning and de-twinning). Nonlinear elastic behavior when the material is relieved from compressive loading is also observed at 1.2% of the total axial strain, resulting in larger plastic strains in compression than in tension. Figure 8 shows that the phenomenological model accurately captures these particular cyclic plastic deformation mechanisms of AZ31B-F magnesium alloy. The Armstrong–Frederick model cannot capture the different strain hardening behavior of MA in tension and compression.

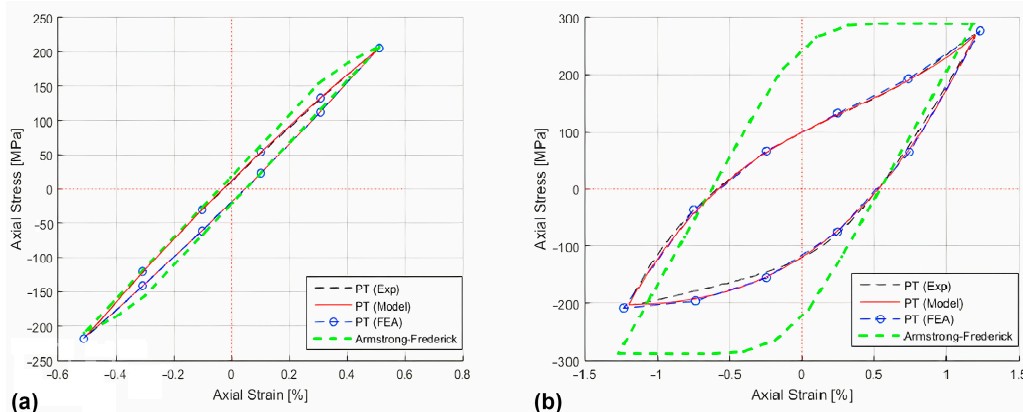

**Figure 8.** Correlation between estimations and experiments for pure axial loading with: (**a**) 0.5% of axial strain amplitude, and (**b**) 1.2% of axial strain amplitude.

### 3.3. Pure Shear (Case PS)

From the pure shear loading, shown in Figure 9, it can be seen that the experimental hysteresis loop in pure torsion is quite symmetric for 0.8% of the strain amplitude. However, for 0.4% of the total shear strain, the hysteresis loop shows asymmetric behavior, which can be inferred from the direction of the first cycle. The shear direction of the first cycle affects the total elastic–plastic deformations. The phenomenological approach captures the hysteresis loops of pure torsional loads with a slight deviation at 0.8% of the shear strain amplitude. The Armstrong–Frederick model fails to capture the maximum stress limits for pure shear loads, because the estimates are much lower than they should be.

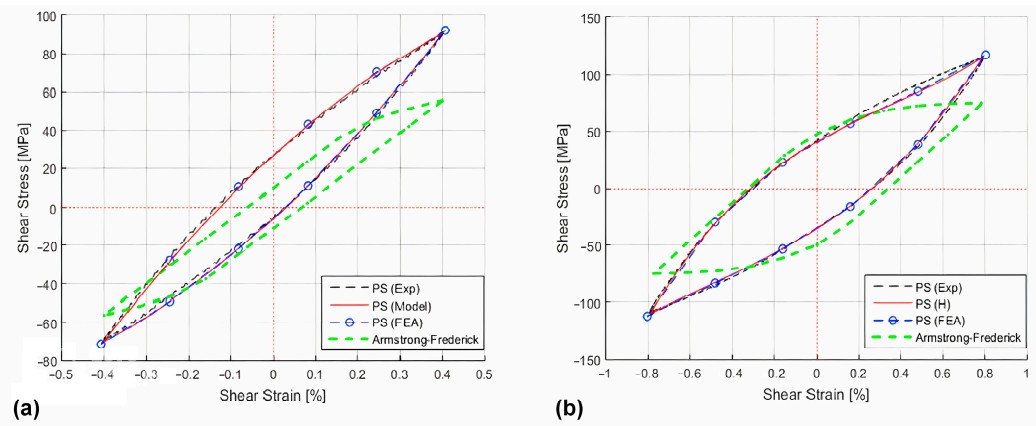

**Figure 9.** Correlation between estimations and experiments for pure shear loading with: (**a**) 0.4% of shear strain amplitude, (**b**) 0.8% of shear strain amplitude.

### 3.4. Proportional Loading with Strain Amplitude Ratio Equal to 30° (Case PP30)

Figure 10 shows a proportional load with a strain amplitude ratio of 30°. In the proportional loading of tension and torsion, the axial and torsional modes are in phase. The axial hysteresis loops in Figure 10 show the main feature of pure cyclic loading (twinning and de-twinning mechanisms). The shear stress–strain curves continue to show symmetrical behavior, as in pure torsion for high strain amplitudes. In this case, the axial strain component is larger than the shear component. Although the axial and inherent plastic deformations determine the deformation behavior, it is observed that they do not significantly affect the shear component.

For proportional loads with a strain amplitude ratio of 30°, the analytical and finite element implementations of the phenomenological approach follow the axial and shear hysteresis loops very well, with only a conservative deviation in the strain hardening behavior of the axial component at a strain magnitude of 1%. For the axial component of

the proportional loads, the Armstrong–Frederick model provides a good approximation of the maximum stress limit in tension for strain sizes of 0.4%, 0.6%, and 1%, but does not capture the maximum stress limit in compression, which becomes more evident at a strain size of 1%. This is due to the fact that magnesium hardening in compression is less than it is in tension. In addition, the plastic strains and back stresses cannot be accurately recorded. This inability becomes more apparent at higher strain values. For the shear component, the Armstrong–Frederick model continues to estimate much lower maximum stress limits than the experimental tests.

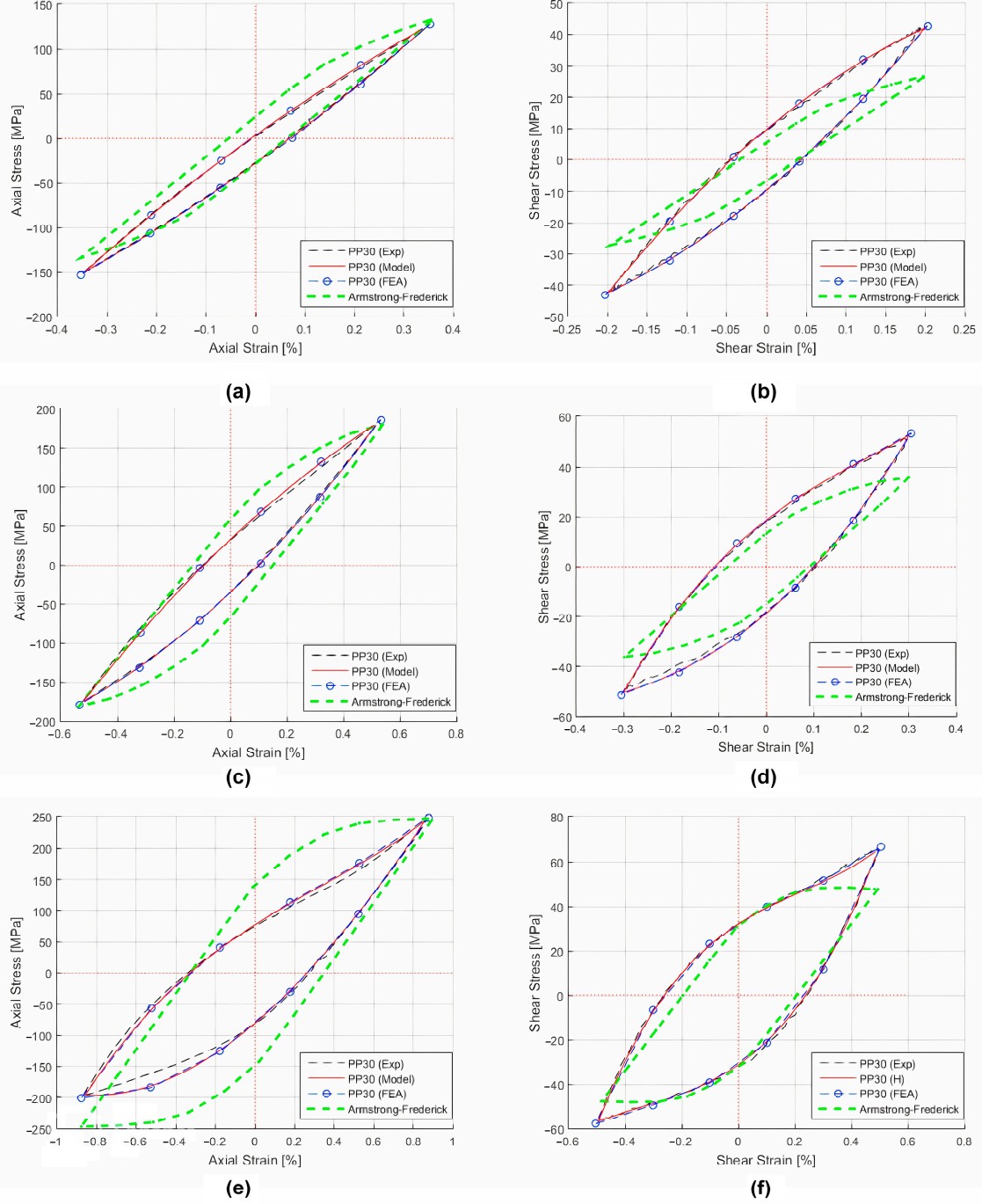

**Figure 10.** Correlation between estimations and experiments for multiaxial proportional loadings with a strain amplitude ratio equal to 30° (**a**,**b**): strain magnitude of 0.4%; (**c**,**d**): strain magnitude of 0.6%; (**e**,**f**): strain magnitude of 1%.

### 3.5. Proportional Loading with Strain Amplitude Ratio Equal to 45° (Case PP45)

Figure 11 shows a biaxial load with a strain amplitude ratio of 45°, which means that the maximum amplitude of the axial and the shear strain are equal. The experimental axial stress–strain curves continue to show an asymmetric pattern. However, the shear component is symmetric at 0.4% and 0.6%, and asymmetric at 1% of the strain amplitudes. This indicates that the torsional mode at 1% of strain amplitude is influenced by plastic deformation mechanisms associated with the axial mode.

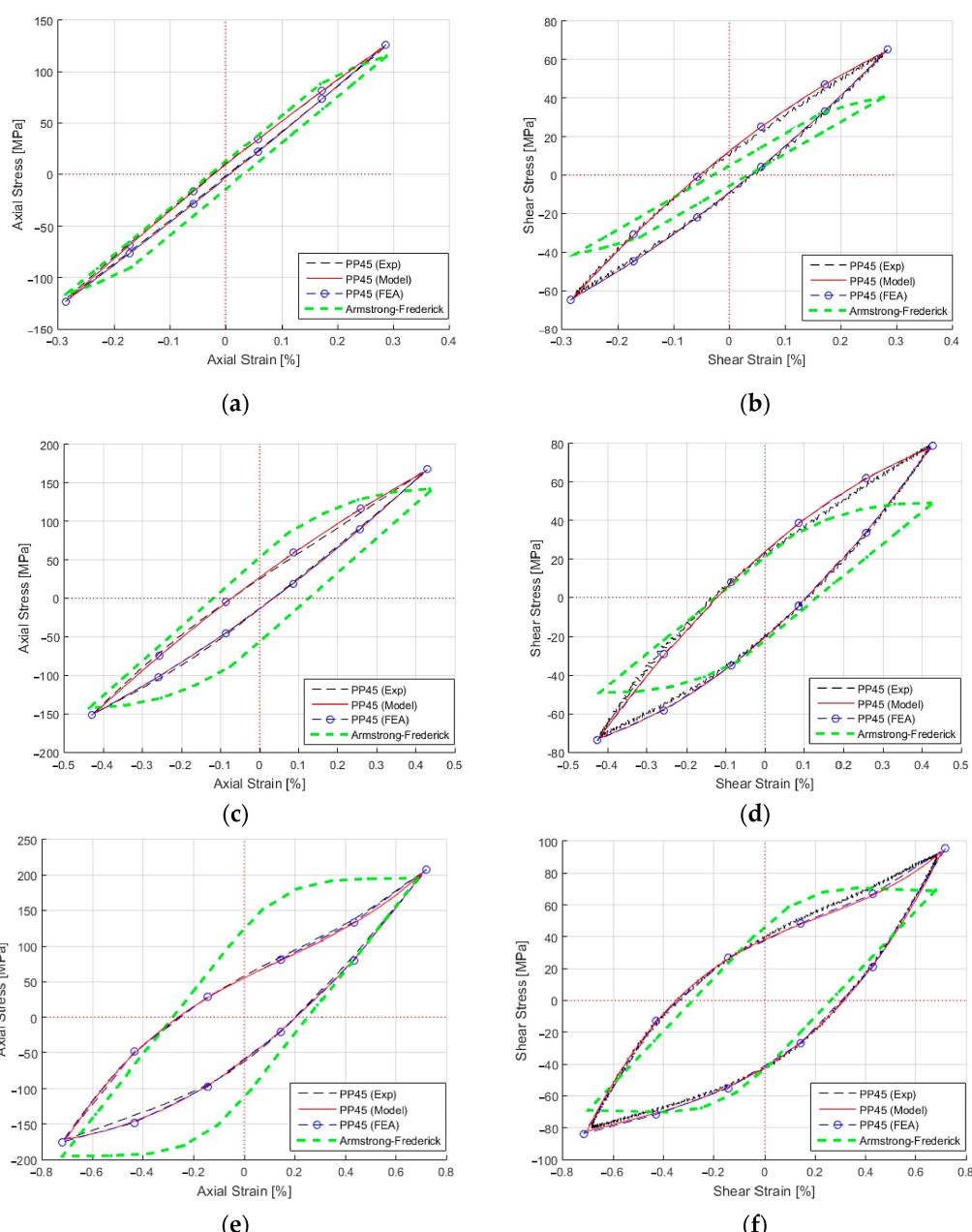

**Figure 11.** Correlation between estimations and experiments for multiaxial proportional loadings with a strain amplitude ratio equal to 45° (**a**,**b**): strain magnitude of 0.4%; (**c**,**d**): strain magnitude of 0.6%; (**e**,**f**): strain magnitude of 1%.

For proportional loads with a strain amplitude ratio of 45°, the analytical and numerical phenomenological approaches accurately describe the stress–strain curves, and only at a strain magnitude of 1% is a deviation observed. For this load path, the Armstrong–Frederick model calculates lower maximum stress limits for the axial and shear components, except for the axial component with a strain magnitude of 1%.

### 3.6. Proportional Loading with Strain Amplitude Ratio Equal to 60° (Case PP60)

Figure 12 shows the results for a proportional load with a strain amplitude ratio of 60°. In this case, the axial and the experimental shear stress–strain curves are not symmetrical. Although the shear strain component is larger than the axial one, the shear behavior seems to be influenced by the axial deformation. For this loading path, the phenomenological approach strictly follows the experiments. However, the Armstrong–Frederick model cannot capture the stress–strain curve of AZ31B-F magnesium alloy.

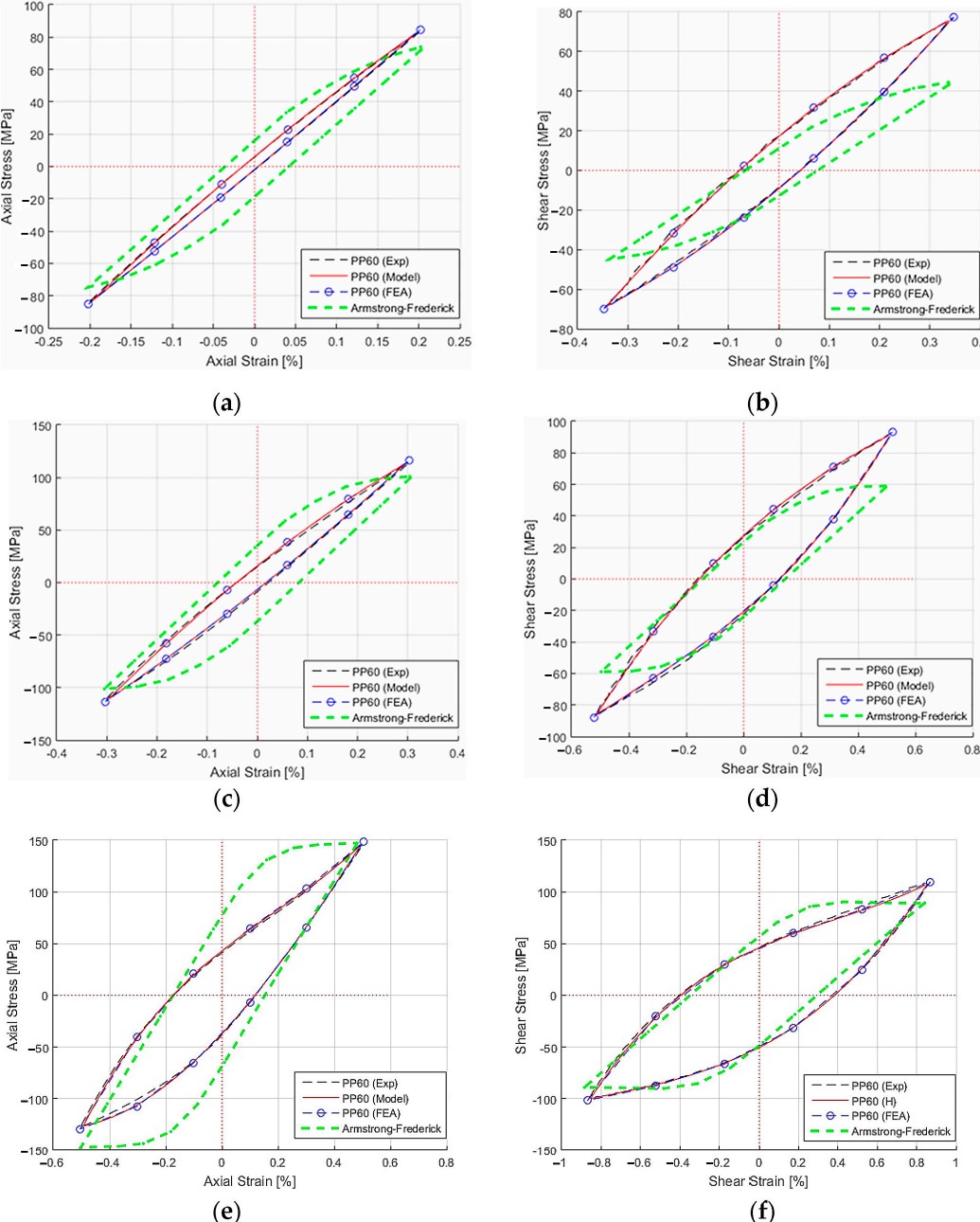

**Figure 12.** Correlation between estimations and experiments for multiaxial proportional loadings with a strain amplitude ratio equal to 60° (**a**,**b**): strain magnitude of 0.4%; (**c**,**d**): strain magnitude of 0.6%; (**e**,**f**): strain magnitude of 1%.

### 3.7. Nonproportional Loading (Case OP45)

Figure 13 shows the nonproportional load path with a strain–amplitude ratio of 45°. The out-of-phase loading is responsible for the rotation of the principal loading axes during

cyclic loading. Figure 13 shows that non-proportional loading produces a different loading pattern in the elastic–plastic cyclic behavior of the material than the proportional loading with a PP45 case.

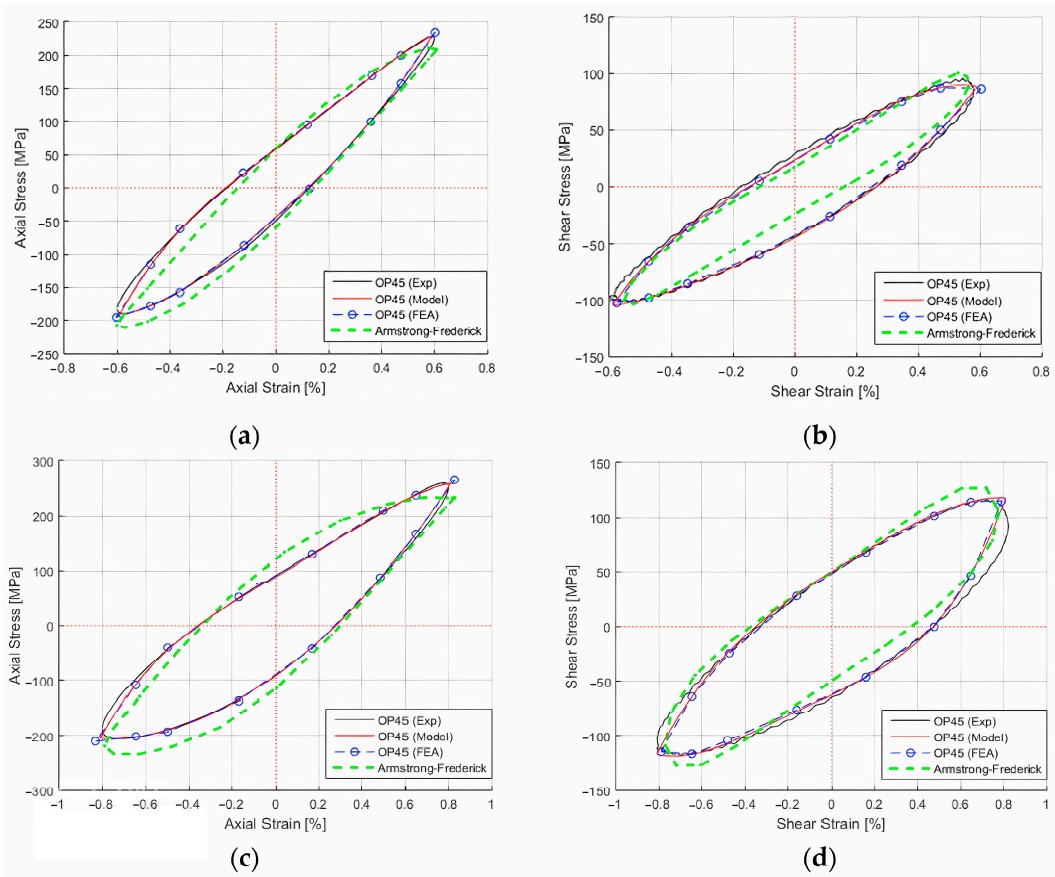

**Figure 13.** Correlation between estimations and experiments for 90° out-of-phase loadings with a strain amplitude ratio equal to 45° (**a,b**): strain magnitude of 0.83% and (**c,d**): strain magnitude of 1.14%.

As with the nonproportional loading, the Armstrong–Frederick model does not follow the experimental results, although it estimates the stresses at maximum strains with acceptable accuracy. For a strain magnitude of 0.83%, the results of the phenomenological model are close to the experiments. However, for a strain magnitude of 1.14%, the model cannot reproduce the shape of the shear hysteresis loop with acceptable accuracy, although it estimates the plastic strains, back stresses, and stresses at maximum total shear strain with very good accuracy. This phenomenological model is based on a polynomial function. Therefore, it is difficult to accurately follow the shape of the shear stress–strain curves of the non-proportional loads. This becomes more evident at higher strain values.

### 3.8. Stress Distribution Analyses

When implementing the phenomenological hypo-strain approach in the UMAT subroutine of Abaqus, the simulations for the different load paths did not encounter any problems with the convergence of the iterations, and very little time was required. Figure 14 shows the simulation results of the phenomenological model implemented in Abaqus/Standard for a tensile-torsional load under the strain amplitude ratio condition of 30° (case PP30). For this biaxial loading, the maximum and minimum axial and shear stresses occur simultaneously during cyclic loading. The Abaqus post-processor was used to study the evolution and redistribution of the elastic–plastic behavior of the AZ31B-F magnesium alloy during cyclic loading. Figure 14a shows a sectional view of the axial stress distribution (σ33) in the specimen. The

maximum axial stress occurs at the extreme end of the specimen. This is a consistent result, since this is the location where the specimen was fixed. Figure 14b shows the distribution of axial stress ($\sigma$33) along the thickness of the specimen during tensile loading at the point where the load reaches its maximum in the stabilized cycle. Similarly, Figure 14c shows the distribution of shear stress ($\sigma$23) along the thickness of the specimen during torsion at the same location. As expected, the results show that the axial loading produces very little stress gradient along the thickness of the specimen, due to its thin wall. During torsional loading, the shear stress varies linearly from approximately 32.5 MPa at the inner diameter to 42 MPa at the outer diameter. The results of the simulations show that the loading path has a strong influence on the stress distribution between the mesh element and the adjacent elements. The simulations of the finite element analysis of the phenomenological model show that they are in agreement with the experimental results.

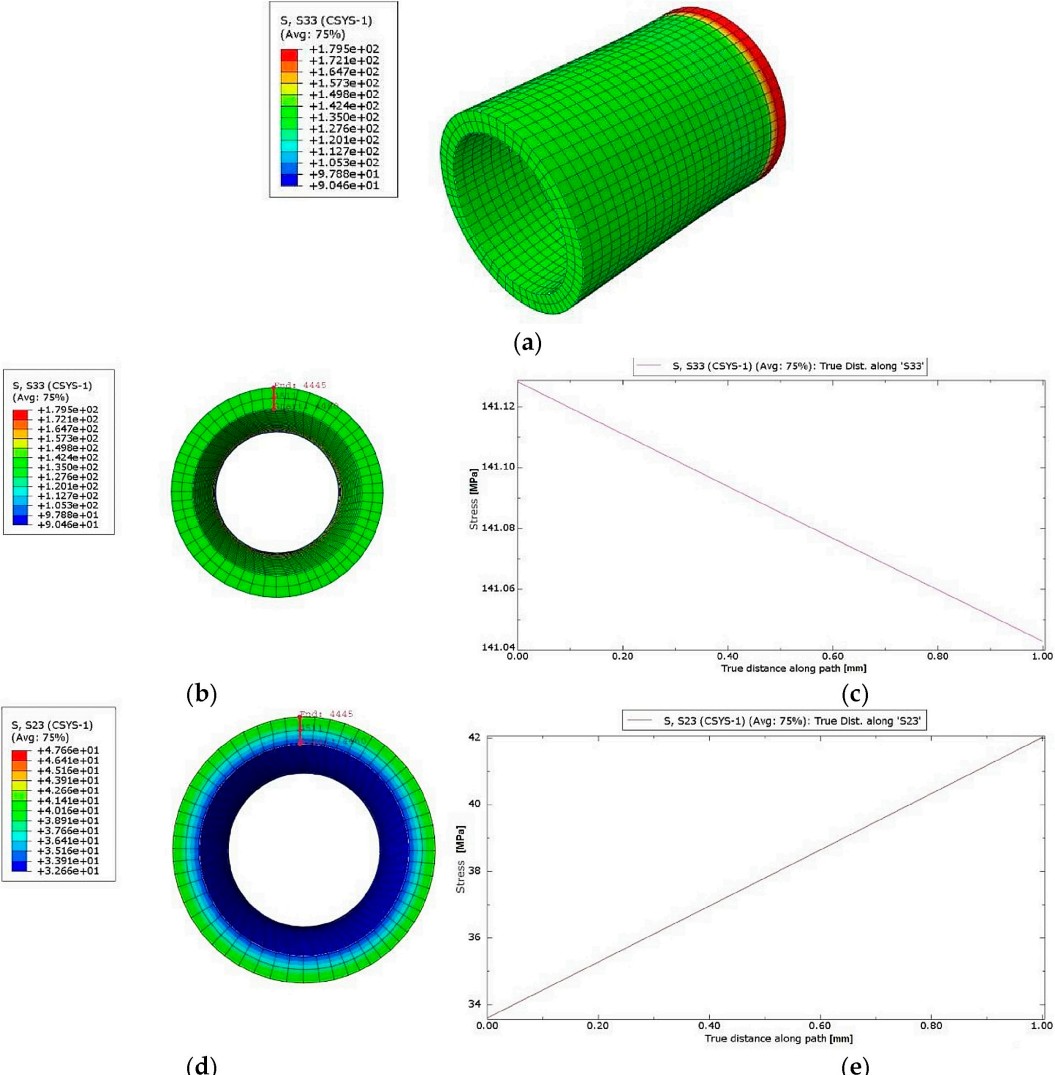

**Figure 14.** Stresses distributions of the phenomenological hypo-strain model implemented in Abaqus, for the tension–torsion loading—Case PP30 (**a**) cutting view image of the specimen; (**b**) and (**c**): axial stress distribution along the thickness of the specimen during tension; (**d**,**e**): shear stress distribution along the thickness of the specimen during torsion.

## 4. Discussion

The results show that the von Mises yield function, hardening rule, and the yield rule used for the Armstrong–Frederick model in Abaqus do not agree with the behavior of

the magnesium alloy. This confirms that this constitutive model is not able to capture the cyclic asymmetry of MA and the deformation effect due to different values of the strain amplitude ratio. Decreasing the strain amplitude ratio increases the axial strain component and decreases the shear component. The results of the multiaxial tests show that the shear component is affected by the axial mechanism of plastic deformation. Therefore, it was concluded that the strain amplitude ratio has an influence on the mechanical behavior of AZ31B-F magnesium alloy. The main advantage of the phenomenological hypo-strain model is the possibility to include the effect of strain amplitude ratio. The stress–strain ranges of the stabilized cycle obtained in the simulations of the phenomenological model can be used in damage models to accurately predict the fatigue life of the wrought alloy MA.

Table 3 shows the average error obtained for the maximum total strains in each loading branch (normal and shear). The minimum average error was 4% and the maximum was 12% for the non-proportional shear component. These results show reasonable correlation between the experiments and the estimates. The nonproportional results are surprising because the phenomenological model includes only the cyclic response of AZ31BF to proportional loading; this result suggests that the sensitivity of AZ31BF to nonproportional hardening is low.

**Table 3.** Correlation between experiments and FEA estimates using the hypo-strain model.

| Type of Loading | Strain Level % | Average Error % |
|---|---|---|
| PT-Pure Axial | 1.2 | 4.0 |
| PS-Pure Shear | 0.8 | 4.0 |
| PP30-Axial | 1.0 | 6.0 |
| PP30-Shear | 1.0 | 4.0 |
| PP45-Axial | 1.0 | 5.0 |
| PP45-Shear | 1.0 | 6.0 |
| PP60-Axial | 1.0 | 4.0 |
| PP60-Shear | 1.0 | 4.0 |
| OP45-Axial | 1.14 | 4.0 |
| OP45-Shear | 1.14 | 12.0 |

Further research is needed to extend the phenomenological hypo-strain model to capture the anisotropic material response of wrought MA considering mean stress effects and to automate the post-processing analysis of Abaqus. In addition, the use of the external Fortran routine developed to update the discrete stiffness matrix in each iteration needs to be improved, since the connection between Abaqus and Fortran is not a simple process. Another limitation is the need to perform a large number of experimental tests to characterize the cyclic response of a given material and implement the corresponding phenomenological model. However, for materials with complex cyclic behavior, such as magnesium alloys, the proposed approach is the one that gives the best results in estimating the stress–strain under multiaxial loading conditions.

## 5. Conclusions

The main objective of this work was to perform a numerical simulation of the relationship between stresses and strains (hysteresis loops) present in the magnesium alloy AZ31BF when subjected to uniaxial and multiaxial loads. These types of simulations performed with constitutive models give poor results due to the complex cyclic response of magnesium alloys, which cannot be estimated without experiments. In this sense, the phenomenological model hypo-strain was used for the numerical simulations of the AZ31BF hysteresis loops, a model previously developed by the authors that takes into account the cyclic and stabilized response of AZ31BF determined in the laboratory under multiaxial loading conditions. The main challenge in this work was to develop a synergistic link between the phenomenological model (hypo-strain) and Abaqus, since the mechanisms available in Abaqus are prepared for constitutive models and not for phenomenological

models. Nevertheless, this linkage was successfully implemented, and several conclusions can be drawn from this work:

- The uniaxial and multiaxial experimental results of the cyclic elastic–plastic behavior of AZ31B-F magnesium alloys were correlated with the phenomenological hypo-strain model and the well-known Armstrong–Frederick model.
- The phenomenological approach was successfully implemented in the commercial finite element program Abaqus/Standard using the UMAT subroutine.
- The Armstrong–Frederick model showed poor estimates, especially for high strain amplitudes and for the axial and shear strain components of multiaxial proportional loads.
- The results were explained by the inability of the model to follow the asymmetric behavior of the magnesium alloy. This conclusion was confirmed by the symmetry of the estimates.
- For uniaxial and multiaxial proportional loads, the phenomenological model followed the cyclic stress–strain curves with very good accuracy. However, a slight deviation was observed for strain values of 1%.
- For the non-proportional loading with a strain magnitude of 1.14%, the phenomenological model fails to capture the shear behavior of the AZ31B-F magnesium alloy, although the model estimates the stresses for the maximum total strain amplitudes, the plastic strains, and the back stresses very well. The stress distributions of the finite element analysis simulations of the phenomenological hypo-strain model are consistent.

The contribution of this work makes it possible to dimension structural and mechanical components made of magnesium alloys using finite element method simulation programs, in particular Abaqus, taking into account the developed cyclic behavior of these alloys due to multiaxial loading conditions. This result will be very useful for fatigue life design of structures and mechanical components made of these alloys, as the current commercial finite element method software is not able to modulate the cyclic behavior of magnesium alloys under multiaxial loading conditions with their intrinsic models.

**Author Contributions:** Conceptualization, V.A., R.M. and L.R.; methodology, R.M.; software, R.M.; validation, M.F., L.R. and V.A.; formal analysis, L.R.; investigation, R.M.; resources, L.R.; data curation, R.M.; writing—original draft preparation, R.M.; writing—review and editing, V.A.; visualization, V.A.; supervision, L.R. All authors have read and agreed to the published version of the manuscript.

**Funding:** This research received no external funding.

**Acknowledgments:** The authors gratefully acknowledge the support from FCT–Fundação para a Ciência e Tecnologia (Portuguese Foundation for Science and Technology), through IDMEC, under LAETA, project UIDB/50022/2020.

**Conflicts of Interest:** The authors declare no conflict of interest.

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
