# Peer review of "Simulation of the Cyclic Stress–Strain Behavior of the Magnesium Alloy AZ31B-F under Multiaxial Loading"

_crystals, doi:10.3390/cryst13060969_

Round 1

Reviewer 1 Report

This work provides a basis for the numerical simulation of cyclic stress in magnesium alloys under multiaxial loading. The work is complex and it has merit. The topic fits the scope of the journal. Please see the detailed comments of the reviewer in the attached annotated PDF file. 

Minor improvements required

Author Response

Dear Reviewer, since your comments were included in the pdf file of the article, I have responded to your comments in the same pdf file that I am sending with these comments. Below are some of the comments I made in the pdf file. All of the suggestions for improvement have been implemented and I believe they have helped improve the quality of the article, thank you.

which is the novelty of this work compared to the state of the art on hypo strain?

  • The work developed here is a pioneering work and the authors are not aware of any similar work in the literature, namely the finite element modeling of the cyclic response of magnesium alloy AZ31BF using the experimental results of tests performed with multiaxial loads. The Hipo-Strain model captures only the stabilized cyclic response of AZ31BF. In this work, we use this cyclic response and, together with the UMAT subroutine, simulate a finite element model in which the stresses are updated with the information from the Hipo-Strain model for AZ31BF - this type of work has never been done before.

Repeated figure

  • This figure is repeated to refer the reader to the section in which it was first described and analyzed. This section presents the stress modeling procedure and the concepts used to update the stress states. Figure 6 shows the various steps to run the program and describes the procedures for each step. The use of the repeated figure facilitates the reader's understanding of the developed framework.

is this a characteristic of this model? why did this model fail to work?

  • The plasticity models available in finite element software are static models based heavily on the von Mises relationship. This relationship is not suitable to account for the response of magnesium alloys under cyclic loading. Therefore, the performance of the A-F model highlights the idea that methods such as the one developed in this work are necessary to incorporate cyclic plasticity models to overcome this limitation.

the discussion is too focused on the A-F model

  • The idea was to focus on the performance of the A-F model to show that methods such as the one developed in this work are necessary to overcome the real-world limitations of commercial finite element method software that do not have cyclic plasticity models. This limitation is particularly important in magnesium alloys.

the models could use symmetry conditions

  • The symmetry constraints are provided by the finite element simulation software, in this case Abaqus. By inserting the phenomenological hypo-strain model into Abaqus via the UMAT subroutine, it is possible to provide Abaqus with the cyclic response of the magnesium alloy. Otherwise, this response would be estimated from the elastic modulus and yield stress. This is a very simplistic approach as it does not take into account the cyclic response of the magnesium alloy resulting from its HCP microstructure, which in turn generates a very peculiar dislocation pattern that has a strong influence on the mechanical response of the magnesium alloy. Therefore, the proposed approach aims to update the material response as a function of the type and level of loading. It is not bound to modulation functionalities and can therefore be used both for simulations with and without symmetry constraints.

can material parameters estimation be improved as well?

  • yes, despite the use of specific values for material parameters, they are always associated with some degree of uncertainty. However, the hypo-strain model has captured the various hysteresis cycles of the cyclic response of the material with a correlation factor R2 greater than 0.95. Therefore, most of the variation in material response will be due to material uncertainty.

How can this work be used in design and in industrial environments? which would be the advantages?

  • The conclusion has been updated with the following: The contribution of this work makes it possible to dimension structural and mechanical components made of magnesium alloys using finite element method simulation programs, in particular Abaqus, taking into account the developed cyclic behavior of these alloys due to multiaxial loading conditions. This result will be very useful for fatigue life design of structures and mechanical components made of these alloys, since the current commercial finite element method software is not able to modulate the cyclic behavior of magnesium alloys under multiaxial loading conditions with their intrinsic models.

Reviewer 2 Report

I have carefully read your manuscript and believe that it can be published in its current form. However, I have a few questions and suggestions regarding your research.

52-55 For me, not very clear what kind of «twins» are talking about. Please describe the terms «twinning» and «de-twinning»

276 What is the final mesh quality. Are two cells across the sample thickness sufficient for this type of simulation?

Can the proposed approach be used for high-cycle fatigue? Will there be a difference for stationary and non-stationary loading cycles?

You are considering only one Armstrong-Frederick plasticity model. Does it make sense to compare other plasticity models such as Ohno-Wang or Prager?

Ultimately, we are interested in the kinetic equation that describes the nonlinear processes of damage accumulation. Have you considered changing the destruction taking into account the proposed model?

Author Response

Dear Reviewer, Thank you for reviewing this article, which I believe has helped improve its quality. Below are the responses to your comments and suggestions for improvement.

52-55 For me, not very clear what kind of «twins» are talking about. Please describe the terms «twinning» and «de-twinning»

  • Twinning is a crystallographic dislocation pattern that occurs in magnesium alloys as a result of cyclic loading. The term twinning is derived from the typical shape of this pattern, which has a typical mirror image about a certain direction. It is a permanent deformation, also known as dislocations. This pattern of displacement of the crystalline structure alters the mechanical properties of the alloy and is essentially characterized by hardening in tension and slight softening in compression, resulting in asymmetric hysteresis cycles. Unlike steel alloys, where the hysteresis cycles are always symmetrical regardless of the level of stress, guaranteeing the same yield strength in tension and compression, these stresses are different in magnesium alloys due to the twin phenomenon, with the tensile yield stress usually higher than the compressive stress. These mechanical properties of magnesium alloys essentially result from their close-packed hexagonal (hcp) microstructure. The de-twinning process results from a load whose orientation leads to deformations that cancel the dislocations previously present, as if the applied load restores the original state of the microstructure shape.

All this information can be found in the references at the end of the sentence that begins at line 48 in the original version of the manuscript.

276 What is the final mesh quality. Are two cells across the sample thickness sufficient for this type of simulation?

  • A convergence study was performed on thin-walled samples used for the simulations to ensure that the number of mesh elements did not affect the results. The final mesh contains 5800 elements with good uniformity. In this study, the shear stress was determined on the outer surface of the specimen. Considering that the shear stress gradient along the thickness is small, only two elements along the thickness were used for the simulations.

Can the proposed approach be used for high-cycle fatigue? Will there be a difference for stationary and non-stationary loading cycles?

  • The proposed approach can be used together with fatigue life estimation models at high cycle fatigue, since the considered phenomenological model estimates the steady-state response of the magnesium alloy. In this sense, the considered model does not take into account the transient region of the material response to cyclic loading, which is more important for low cycle fatigue conditions.

You are considering only one Armstrong-Frederick plasticity model. Does it make sense to compare other plasticity models such as Ohno-Wang or Prager?

  • The idea of this study was first to implement the phenomenological model in the finite element simulation program Abaqus and to compare the estimates of the phenomenological model with the models offered by Abaqus. To this end, we selected the Armstrong-Frederick model for this comparison. Another objective was to show that the plasticity models offered by Abaqus do not represent the cyclic response of magnesium alloys and that by including the phenomenological model it is possible to overcome this limitation. On the other hand, the given plasticity models (Ohno-Wang and Prager) are very important and it would be interesting to correlate the estimates between the given models and the proposed approach. This correlation could be done in future work.

Ultimately, we are interested in the kinetic equation that describes the nonlinear processes of damage accumulation. Have you considered changing the destruction taking into account the proposed model?

  • This is a very interesting question, and indeed damage assessment in its simplest form is derived from the ratio between the number of cycles loaded at a given amplitude and the number of cycles to failure at the same amplitude (Palmgren-Miner rule). The number of cycles to failure is determined by a SN curve and by the magnitude of the loading amplitude. It is extremely important to have the correct loading amplitude in order to estimate the correct number of cycles and thus obtain a better estimate of the cumulative damage. In this sense, the proposed approach fulfills this improvement, since it allows the estimation of the developed stresses in finite element simulations and, consequently, a more accurate evaluation of the cumulative damage in these simulations. The fact that the loading stresses are closer to those verified in reality will also allow the development of nonlinear damage accumulation models in a more accurate way.

Reviewer 3 Report

The article uses a numerical model to describe the behavior of magnesium under cyclical loads. The article is well written. It smoothly guides the reader through the research methodology and its results. 

However, there are shortcomings:

1. Lines 42-47 and 48-53 are repeated.

2. Line 54: The next paragraph begins with the same words. I suggest modifying it.

3. Figure 2a: The figure is poor quality.

The authors refer in the introduction to microstructural factors that affect the obtained results. However, they do not analyze these factors in the discussions and conclusions. More expansion is required in this regard.

Author Response

Dear Reviewer, Thank you for reviewing this article, which I believe has helped improve its quality. Below are the responses to your comments and suggestions for improvement.

  1. Lines 42-47 and 48-53 are repeated.
  • The repeated paragraph has been removed.
  1. Line 54: The next paragraph begins with the same words. I suggest modifying it.
  • The manuscript has been improved according to the indicated instructions.

  1. Figure 2a: The figure is poor quality.
  • The quality of the figures has been improved.

The authors refer in the introduction to microstructural factors that affect the obtained results. However, they do not analyze these factors in the discussions and conclusions. More expansion is required in this regard.

  • This question is relevant. The microstructural factors arising from the hexagonal (hcp) microstructure and affecting the mechanical behavior of the magnesium alloy are implicitly analyzed by the phenomenological model, i.e., the phenomenological model contains a representation of the stress-strain response under multiaxial loading, which in turn arises from the particular microstructural displacements (twinning and de-twinning) that occur in magnesium alloys. However, the main focus of this work is to integrate the phenomenological model into a commercial finite element analysis software (Abaqus) and to verify that the plasticity models available in commercial finite element software are neither prepared nor have tools to simulate the stress state for magnesium alloy structures or mechanical components. In this work it was possible to prove this and present an alternative for this limitation.

Round 2

Reviewer 1 Report

Thank you for the improvements

Minor review required

Reviewer 3 Report

I have no more objections,